# Expertise-dependent differences in mental representation metrics of pas de bourrée

Pia Wittenbrink[1], Mira Janzen[1], Antonia Jennert[1], Benjamin Strenge[1,2]*

1 Faculty of Psychology and Sports Science, Bielefeld University, Bielefeld, Germany, 2 Neurocognition and Action – Biomechanics Research Group, Center for Cognitive Interaction Technology (CITEC), Bielefeld University, Bielefeld, Germany

* benjamin.strenge@uni-bielefeld.de

**Data Availability Statement:** All relevant data are within the manuscript and its Supporting information files.

**Funding:** We acknowledge support for the publication costs by the Open Access Publication

## Abstract

Precise movement control is of prime importance in almost every kind of sport and greatly influences an athlete's performance. In dancing not only motor but also cognitive skills, e.g. in the form of memorized representational structures, are essential components of the performance. This study investigated different metrics related to the long-term memory of ballet dancers with different skill levels regarding the pas de bourrée using the structural-dimensional analysis of mental representations (SDA-M) method. To this end, the Correct Action Selection Probability Analysis (CASPA) algorithm, a recent SDA-M extension that predicts the individual probabilities of correct action selections within a movement sequence, has been applied in the context of dancing for the first time. Significant positive correlations were found between participants' degree of expertise and the proximity of their mental representation structure to the ideal reference structure, as well as between the degree of expertise and the probability of correct action selection within the movement sequence estimated by CASPA. The results indicate that increased training experience in ballet dancing is not only associated with functionally better structured mental representations of the movement sequence but also with a higher probability of correct action selection. These findings provide further evidence for SDA-M with CASPA as an auspicious tool for individualized task-related memory assessments and diagnostics in different action sequences, e.g. as the basis for mental training.

## Introduction

Mental representations comprise sensory effects and movement structures and therefore play a crucial role in the control and organization of actions [1, 2]. They enable versatile, task-specific, and cognitive functions, such as anticipation or the consideration of alternative courses of actions, e.g., in order to successfully perform an aiming movement [2]. This is accomplished by storing the cognitive units of complex movements on the level of mental representation. These units are called basic action concepts (BACs) [3]. Each individual BAC contains a typical set of sensory and functional features [2]. Hence, BACs are defined as "basic units of knowledge, accessible for mental control in volitional acts, as sensorimotor representations of movement effects, and also as cognitive tools for the execution of actions like complex movement tasks" (p. 351) [2]. In this context, mental representation structures have been described as

Fund of Bielefeld University and the Deutsche Forschungsgemeinschaft (DFG). The funders had no role in study design, data collection and analysis, decision to publish, or preparation of the manuscript.

**Competing interests:** The authors have declared that no competing interests exist.

"integrated networks of basic action concepts (BACs) which are mapped for both individuals and social groups by giving information about relational structures in a given set of concepts related to goal-directed actions" (p. 6) [1]. Thus, a functional organization of the BACs and mental representations are important regarding movement control and correct movement execution.

In order to investigate cognitive structures in long-term memory, the structural-dimensional analysis of mental representation (SDA-M) method was developed [4]. This approach requires the participants to make knowledge-based decisions in a so-called "split procedure", from which individual clusters of mental representations are computed and visualized as a dendrogram [5]. SDA-M has already been applied in a variety of studies in a sports context [5–7]. For instance, the modification of mental representations of novice golfers has been analyzed as a result of four different training intervention [6]. Besides, mental representational structures have also been examined using a throwing technique of judokas who were current or former members of the German national judo team. In previous research, several authors compared mental representations of novices versus experts by means of SDA-M. These studies were conducted in the disciplines like climbing, canoeing, soccer, volleyball, tennis, overhead throwing and windsurfing [8–14]. Results of these studies showed significant differences in mental representations depending on the level of expertise. Thus, BACs were clustered in more functional ways among the experts than among the novices. This means the higher the level of expertise, the more the biomechanical structure of the movement is reflected and the more distinctive the mental representations of the athletes are [2]. Furthermore, the cluster solutions within the expert groups tended to vary less.

Complementary to these established applications, SDA-M has recently been extended by the Correct Action Selection Probability Analysis (CASPA) algorithm, which utilizes approaches of the computational cognitive architecture "Adaptive Control of Thought—Rational" (ACT-R) for estimating the probability that a specific participant would select a correct follow-up action after successfully executing a given part of an action sequence [15]. The mean of these individual probability estimates can be interpreted as an overall estimate of the expected probability of correct action selection in a randomly chosen situation within the movement sequence and allows for a more fine-grained assessment of mental representation structures [16]. CASPA has been used in a first study in the context of sports to assess practitioners' mental representation structures regarding a choreographed movement sequence in karate (the Kanku-dai kata) [7]. It was found that the performance predictions generated by CASPA correlated significantly with the karatekas formal expertise and the likelihood of errors in action execution [7].

Since mental representations are essential in learning complex movement sequences, as well as in improving and fine adjusting of motion sequences with regards to their aesthetics, they also have an essential meaning in dancing [2]. Consequently, a dancer needs not only physical control but also cognitive skills [17]. A number of studies have already investigated cognitive skills in dancing. Especially, investigations measuring the brain activity of dancers using fMRI [18–21] and EEG [22, 23], as well as measuring the cognitive performance in dance experts and novices [24, 25], dominated this area of research. Based on those previously mentioned studies, the importance of mental representations in the performance, learning, observation, and recall of movements is clearly evident. Therefore, the quality of these mental representations can be classified as an essential characteristic of a dancer's level of expertise [2].

Nonetheless, only a single study investigated movement sequences of ballet using SDA-M to examine mental representations in long-term memory [2]. In this investigation, two basic movement sequences of classical ballet were analyzed, namely the Pirouette en dehors and the

Pas assemblé. To this end, male and female participants were divided into different groups depending on their level of expertise and their training experience in classical ballet. The results of the SDA-M demonstrated that participants with different levels of expertise showed movement-specific differences in their mental representations in long-term memory. Only the experts were able to categorize the movement into functional sequences and thus put them into a functional structure [2]. Despite these significant findings in this particular study, the broad range of study participants' experiences limits the study's power. For instance, the study group of amateurs in the Pas assemblé includes participants with a difference in experience of up to 19.75 years. Another consideration that needs to be taken in account is that participants had the opportunity to stand up and perform the movement during the split procedure. This gave them the opportunity to carefully consider and question whether two shown BACs were functionally related to each other or if they were not. As a result, there is the possibility that decisions are no longer made intuitively but rather analytically. This is emphasized by the time the participants took to complete the split procedure (Pas assemblé: 15 minutes to one hour for nine BACs; Pirouette en dehors: 30 minutes to two hours for 16 BACs).

The goals of the present study were to demonstrate the applicability of CASPA in the context of dancing and to assess mental representation structures in ballet in more detail to expand the knowledge about the impact of expertise. To this end, the mental representation structures regarding the Pas de bourree movement sequence of participants with different levels of expertise were analyzed using classic SDA-M metrics and the more recent CASPA approach. Thus, this study is the first one using the CASPA approach to analyse a movement sequence in classical dancing, which enables the prediction of human errors to facilitate assessment and training processes. Based on the current state of knowledge, we expected positive correlations between participants' level of expertise and the degree of invariance of their cluster solution to an ideal reference structure, as well as between the level of expertise and the probabilities of correct action selection estimated by CASPA.

## Materials and methods

### The pas de bourrée, its variations and basic action concepts

The pas de bourrée sequence represents a common element in classical ballet [26]. It is a fluid movement, which can be performed in different directions and variations. Examples of variations are the addition or omission of rotations and foot changes. Particularly precise footwork is of great importance in order to execute the movement sequence correctly [27]. The pas de bourrée is introduced for beginners and modified as expertise increases, including the aforementioned variations [26]. One variation of the pas de bourrée is the pas de bourrée piqué, which is examined in the present study. It involves the doubled placement of the working leg in the retiré position, in which the toe is put below the knee (see $BAC_4$ and $BAC_6$ in Fig 1). Additionally, a very sharp movement character is typical for the pas de bourrée piqué [27]. To improve the readability, the term of pas de bourrée piqué is abbreviated to pas de bourrée in the following. In general, the pas de bourrée is a symmetrical and mirrored movement consisting of nine BACs (see Fig 1). These BACs were defined and phrased in line with the standard recommendations for researchers and practitioners from [3] based on movement descriptions from standard references on classical dance training [26, 27], as well as complete and detailed descriptions of the movement sequence by an experienced ballet teacher. These descriptions were broken down into distinct key points to derive BACs for the experiment. The comparison of $BAC_1$ and $BAC_9$, $BAC_2$ and $BAC_8$, $BAC_3$ and $BAC_7$, as well as $BAC_4$ and $BAC_6$ clarifies that those BACs only differ in minor details. For example, $BAC_4$ and $BAC_6$ each have one leg in the Retiré position, but affecting the right leg in $BAC_4$ and the left leg in $BAC_6$.

| BAC | Image | Description |
|---|---|---|
| 1 |  | Left foot in front |
| 2 |  | Left leg bends, right foot is put on |
| 3 |  | Half top stand on ball of foot, left foot in front |
| 4 |  | Left foot is applied to the right knee |
| 5 |  | Step to the side, stand on the balls of your feet |
| 6 |  | Right foot is applied to the left knee |
| 7 |  | Half top, stand on ball of foot, right foot in front |
| 8 |  | Right leg bends, right foot is put on |
| 9 |  | Right foot in front |

**Fig 1. Basic action concepts of the pas de bourrée.**

## The SDA-M method

The "structural-dimensional analysis of mental representations" (SDA-M) method is used as an experimental method for collecting and analyzing psychometric data about the structure of mental representations [1]. All SDA-M procedures in this study were performed using the software QSplit SDA-M Suite v1.7 for Windows. First, a split procedure is performed using the predefined BACs. To this end, participants complete a multiple sorting task with pairwise comparisons in which they have to decide whether the two BACs, which are presented on the screen, are directly associated during execution of the movement sequence [7]. Based on the results of the procedure, the software provides a distance scaling of the BACs [1]. These distance values serve as an input for computing an estimated probability that the participant would be able to select correct actions for transitioning from one BAC to the next when executing the movement sequence [16]. Subsequently, cluster analysis is used to determine groupings of BACs as a hierarchical structure and a corresponding individual dendrogram is obtained. These dendrograms visualize the individual cluster solutions in form of a hierarchical and tree-like structure. Finally, the cluster solutions are analyzed for invariance within or between groups.

## Participants

A power analysis with SPSS Version 29.0.0 estimated that a minimum sample size of 16 participants would have been necessary to determine significant Spearman correlation values of $\rho = .7$ or higher, which are commonly interpreted as "strong" [28–30], with at least 80% chance. In order to establish three perfectly balanced expertise groups of six participants each, a total of 18 participants aged between 19 and 33 years were recruited. Four of them were male (22.22%) and 14 were female (77.77%). The detailed participant demography data is provided as S2 File. The participants were divided into three different expertise groups based on their total number of ballet training sessions, which had been estimated by calculating the product of their ballet training frequency per week, their ballet training experience in years, and the factor 50 representing the number of weeks per year. Each group consisted of six participants. The participants who were categorized as novices did not have any experience in dancing (average age: 22.66 years). Participants with a score between 50 and 450 were classified as beginners (average age: 26.33 years; between 1 and 2 years of ballet training once or twice per week). Those with a score of 900 or higher were aggregated as the advanced group (average age: 24.5 years; more than 4 years of ballet training 2 to 4 times per week). For example, a person who had practiced ballet twice a week for 14 years would have received a score of 14*50*2 = 1400 and be assigned to the advanced group. All calculated experience scores can be found in S1 File.

## Execution

The study was approved by the Ethics Committee of Bielefeld University (EUB). Participants were recruited in March 2021 at the campus of Bielefeld University and from a ballet school in Bielefeld (Germany), gave written informed consent to participate in the study and filled out a questionnaire regarding their age, main sport discipline, training experience related to that discipline in years, and the number of training sessions completed per week. Due to the COVID-19 pandemic, all data collection was conducted online via the video portal Zoom. Therefore, during data collection the experimenter could identify individual participants, but all their data was immediately pseudonymized to prevent subsequent identification of individual participants. The experimental procedure can be divided into four phases.

**Phase 1: SDA-M introduction.** The split procedure was demonstrated using an instructional video included in the QSplit software. The instructions given within the video were the following:

*"The software shows representations of two action steps. You are to judge whether these action steps are sequentially "directly associated" or not during task execution, i.e., whether they are executed immediately before or after each other. It does not matter which action step is shown on the left or right side of the screen."*

Afterwards, the execution was illustrated through the simple example of toasting a slice of bread. Subsequently, the participants had the opportunity to ask follow-up questions to the experimenter. Finally, participants confirmed general comprehension of the instructions.

**Phase 2: Video sequence of the movement.** Participants watched a video sequence of a dancer performing the Pas de bourée. This video was shot from behind, as were the BACs. The frequency of playing the video was related to the expertise level of the participants. Based on training frequency and training experience, it could be assumed that the beginners and the advanced participants had already observed and performed the pas de bourrée numerous times during practice. Therefore, novices were allowed to watch the video sequence three times, beginners and advanced participants watched it twice. The purpose was to level the playing field between experimental groups to some degree without inducing physical practice.

**Phase 3: Presentation of basic action concepts.** The nine BACs were presented to the participants in a random order before the start of the split procedure. This enabled the participants to assess the level of abstraction of the BAC representations.

**Phase 4: SDA-M split procedure.** Afterwards, the participants performed the split procedure, which comprised the nine BACs of the pas de bourrée and corresponding text descriptions (see Fig 1). They stated each decision verbally (via microphone) and the experimenter entered it into the QSplit software.

## Data analysis

The ideal mental representation structure was created manually by performing a split procedure in which exactly those BACs that are executed immediately before and after one another within the movement sequence were associated to each other. The resulting structure consists of four individual clusters (see Fig 2a). The first cluster includes $BAC_1$ and $BAC_2$, the second cluster contains $BAC_3$ and $BAC_4$, the third cluster consists of $BAC_5$ and $BAC_6$, and the fourth cluster comprises $BAC_7$, $BAC_8$ and $BAC_9$. This structure served as a reference for assessing each group's mean result, as well as the results of the individual participants. All analyses were based on a significance level of $\alpha = .05$. The horizontal red line in an SDA-M dendrogram from the QSplit software represents the critical Euclidean distance value ($d_{crit} = 3.48957$ for a significance level of $\alpha = .05$). Accordingly, all BACs that form structures below this critical value are grouped into clusters. From a memory research point of view, the lower the distance, the closer the BACs are represented together in long-term memory [5]. A low Euclidean distance is graphically expressed in a low projection on the vertical axis in the dendrogram [1]. An invariance analysis of the cluster solutions was performed using Lander's λ invariance measure as described by [3]. Consequently, cluster solutions were classified as invariant if the value of λ was higher than $\lambda_{crit} = .68$. Furthermore, Correct Action Selection Probability Analysis (CASPA) was conducted using the same SDA-M data. The $CASPA_m$ metric then represented the arithmetic mean of all estimated probabilities of successful action selections over the entire movement sequence based on a given participant's SDA-M data. This value can also be

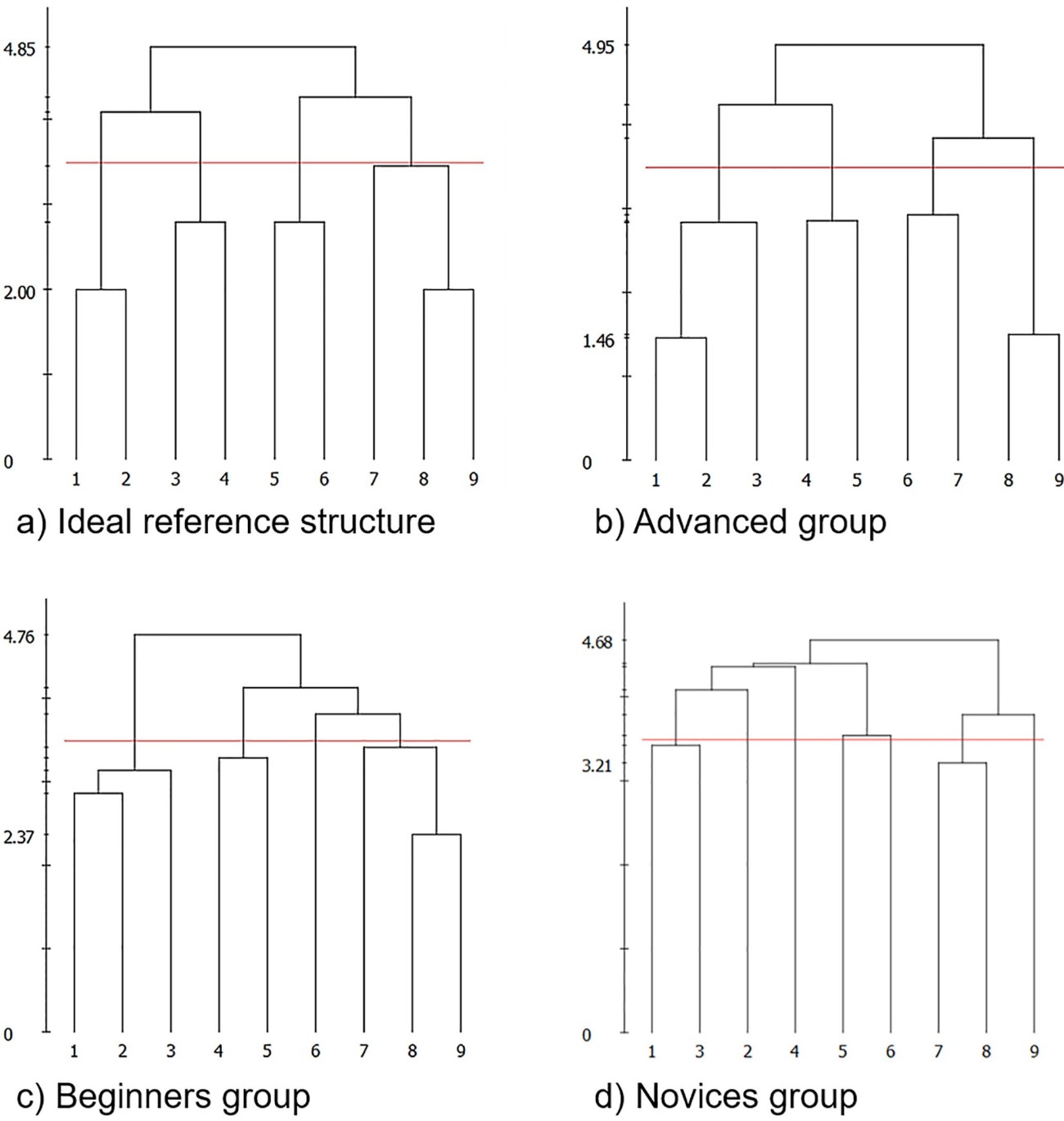

**Fig 2. Dendrograms of (a) the ideal reference structure, (b) the advanced group, (c) the beginners group, and (d) the novices group.** The horizontal red line represents the critical Euclidean distance value ($d_{crit}$ = 3.48957 for a significance level of $\alpha$ = .05). The average pairwise distance of all BACs connected in a cluster under this line corresponds to a correlation between the respective BAC vectors.

interpreted as an overall estimate of the expected probability of correct action selection in a randomly chosen situation within the movement sequence [7]. Participants' training experience was then correlated with their individual $\lambda$ and $CASPA_m$ values. The respective raw data is provided as S1 File.

## Results

Fig 2 shows the dendrograms of a) the ideal reference structure, b) the advanced group, c) the beginner group and d) the novice group. These dendrograms illustrate that with a higher level of expertise, the proximity to the ideal cluster solution generally increases. Consequently, the mental representation structure of the advanced group (n = 6) is closest to the ideal representation structure (see Table 1). Nonetheless, the cluster solution of the advanced group slightly differs from the cluster solution of the reference structure ($\lambda = 0.502118$) and thus can not be classified as invariant. The mental representation structure of the advanced group is divided into four individual clusters, comparable to the ideal representation structure. However, there is a difference between the reference structure and the advanced participants' group mean structure in the assignment of the BACs to the individual clusters. In the advanced group, the first cluster includes $BAC_1$, $BAC_2$ and $BAC_3$, the second cluster includes $BAC_8$ and $BAC_9$, the third cluster includes $BAC_4$ and $BAC_5$, and the fourth cluster includes $BAC_6$ and $BAC_7$ (see Fig 2b). There are additional differences in mental representational structures among the individual advanced group members, with the mental representations of three participants being invariant ($\lambda = 0.70368$; $\lambda = 1$; $\lambda = 1$) and the mental representations of three participants being variant from the reference structure ($\lambda = 0.466287$; $\lambda = 0.453715$; $\lambda = 0.456053$). In the group of beginners (n = 6), their mental representation structure is variant compared to the ideal reference structure ($\lambda = 0.453715$) and compared to the representation structure of the advanced group ($\lambda = 0.515156$). Although the advanced group and the beginners showed the same cluster solution for the first part of the movement, a difference occurred in the second part of the movement. While the advanced group summarized $BAC_6$ and $BAC_7$, as well as $BAC_8$ and $BAC_9$ into a cluster, the beginners grouped $BAC_7$, $BAC_8$ and $BAC_9$ into one cluster and created an additional cluster that consisted of $BAC_6$ (see Fig 2c). The biggest deviation compared to the reference structure was found for the group of the novices (n = 6; $\lambda = 0.344021$). The dendrogram of the group of novices (see Fig 2d) shows that almost all BACs were isolated in individual clusters. Only $BAC_7$ and $BAC_8$, as well as $BAC_1$ and $BAC_3$, were clustered together below the critical value of the Euclidean distance.

An overview of Lander's $\lambda$ invariance values of groups' mental representation structures in relation to the ideal reference structure is shown in Table 1. Additionally, a highly significant positive correlation between the level of expertise and the proximity to the reference structure was found (Spearman's $\rho = .665$, $p = 0.003$; see Table 2).

**Table 1. Lander's $\lambda$ invariance values of groups' mental representation structures in relation to the ideal reference structure and average $CASPA_m$ values for participants from each group.**

|  | Novices | Beginners | Advanced |
|---|---|---|---|
| **Lander's $\lambda$** | 0.34 | 0.45 | 0.5 |
| **$CASPA_m$** | 0.217 | 0.364 | 0.82 |

**Table 2. Correlation of expertise with Lander's $\lambda$ invariance and $CASPA_m$ values.**

|  | Lander's $\lambda$ | $CASPA_m$ |
|---|---|---|
| **Degree of expertise** (Training experience * Training frequency) | .665** | .836*** |

** $p < .01$,

*** $p < .001$

The estimated probability of correct selection of actions also increased with the degree of expertise (see Table 1). Thus, a $CASPA_m$ value of 0.819509 was determined for the advanced group and a value of 0.364204 for the beginners. The lowest $CASPA_m$ value of 0.217105 was determined for the novices. Furthermore, as shown in Table 2, there was a highly significant and strong positive correlation between the proximity to the reference structure and the predicted probability of correct action selection (Spearman's $\rho$ = .836, $p < 0.001$).

## Discussion

This study, in line with the current body of literature, indicates that with growing expertise the representation structure of the movement corresponds increasingly to its topological spatio-temporal structure and thereby possesses intrinsic spatiotemporal properties [2]. In the context of sport, this offers the advantage of an enhanced motor control of movement [2]. Accordingly, we found major differences in the mental representation in long-term memory of advanced ballet dancers, beginning ballet dancers and novices for the Pas de bourée. As the level of expertise increased, the proximity of the individual cluster solution to the ideal reference structure increases as well. Thus, the BACs were clustered in more functional ways among the advanced ballet dancers compared to beginners and novices. This confirms the proposition that mental representations have an essential meaning in dancing [2]. Nevertheless, the present study found some variant mental representations in three of the advanced ballet dancers compared to the ideal cluster solution. This could partially be attributed to complicating properties of the symmetrical and mirrored movement, i.e., the respective BACs only differ from one another in minor details. Additionally, although the advanced participants were experienced ballet dancers, they were not professionals.

According to our knowledge, this study is the second published one using the $CASPA_m$ as an advanced alternative for assessing expertise based on SDA-M data in sports, as well as the first one to do so in the context of dancing. The utilization of CASPA automatizes the assessment of memory structures and enables the prediction of probable errors in movement sequences. In this study, a significant and strong positive correlation was found between expertise and the predicted probability of correct action selection (Spearman's $\rho$ = .836, $p < 0.001$). This is especially relevant in the context of the analysis of the pas de bourrée, because this movement sequence represents a basic element in classical ballet [26]. Nevertheless, large differences of $CASPA_m$ between the beginners and advanced group were found in this study. Hence, these results may add great value in ballet training by generating individualized performance predictions for a movement sequence especially for novices and beginners of dancing. This individual knowledge facilitates the learning of new movement sequences by identifying and eliminating the respective errors in execution.

Some limitations of the study were caused by the COVID-19 pandemic. It was neither possible to recruit a larger sample size nor to recruit any professional ballet dancers to participate in this study. Although the participant number exceeded the minimum sample size estimated by power analysis, future larger studies could facilitate the determination of more reliable estimates of the actual population values. Another limitation was the execution of the split procedures via online video conferencing. This resulted in difficulties playing the video sequence due to a poor internet connection in a few cases. From a methodological point of view, the inherent properties of a correlation study must be taken into account when interpreting the study results, such as that it only studies the relationship of variables but does not investigate causality.

Notwithstanding these limitations, the current study provided valuable insights for practical applications. Utilizing the SDA-M method and its recent algorithmic extension CASPA, new

types of individualized training programs can be designed, which focus on one's particular deficits. For instance, expertise-related instructions can be given and individual technical training routines in high-performance sports can be optimized through automatized assessment and corresponding mental training. Furthermore, future research should investigate sex differences, as well as other, less symmetrical movement sequences of classical ballet. Future studies may include an additional group of participants consisting of professional ballet dancers to generate a larger sample size and explicitly differentiate the mental representations of professional ballet dancers from dancers with a merely advanced expertise level. Besides high-performance sports, it is worth investigating the use of SDA-M and CASPA in the health sector. For instance, representations of motor actions in stroke patients have already been analyzed using SDA-M [31]. In this context, advanced analysis approaches as used in the present study could provide an improvement in the therapy and rehabilitation process due to better assessment and individualization.

## Conclusion

The methodology and results of the present study not only corroborated but also extended the current state of research regarding the study of mental representations by applying the SDA-M-based CASPA algorithm in classical dancing. CASPA served as a tool for assessing memory structures to predict probable errors in action sequences and provided additional knowledge about the ramifications of expertise on mental representations and performance in dancing. Accordingly, the results of the present study clearly demonstrated that practice leads to functional adaptations in the representation structure of the pas de bourrée, which was reflected by higher estimated probabilities of correct action selections within the movement sequence. Consequently, SDA-M analyses and its recent algorithmic extension CASPA can be used to assess the expertise and identify deficits of athletes regarding their mental representation structures in order to improve them through specific training, support therapy and rehabilitation processes. To implement and evaluate the latter proposal, further research is required.

## Supporting information

**S1 File. Raw data.** Includes the study raw data, including participants' calculated training experience scores, Lander's lamda value and CASPA value.
(XLSX)

**S2 File. Overview of the subjects.** Includes more detailed information regarding gender, age and training experience of the subjects.
(PDF)

## Acknowledgments

We would like to thank Thomas Schack for his support and promotion of the study.

## Author Contributions

**Conceptualization:** Benjamin Strenge.

**Data curation:** Pia Wittenbrink, Mira Janzen, Antonia Jennert.

**Formal analysis:** Pia Wittenbrink, Mira Janzen, Antonia Jennert, Benjamin Strenge.

**Investigation:** Pia Wittenbrink, Mira Janzen, Antonia Jennert.

**Project administration:** Benjamin Strenge.

**Resources:** Pia Wittenbrink, Mira Janzen, Antonia Jennert, Benjamin Strenge.

**Software:** Benjamin Strenge.

**Supervision:** Benjamin Strenge.

**Validation:** Benjamin Strenge.

**Visualization:** Pia Wittenbrink, Mira Janzen, Antonia Jennert.

**Writing – original draft:** Pia Wittenbrink, Mira Janzen, Antonia Jennert.

**Writing – review & editing:** Pia Wittenbrink, Benjamin Strenge.

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
