## [Decision Letter · Decision Letter 0]

12 Jul 2023

PONE-D-23-11698Expertise-Dependent Differences in Mental Representation Metrics of Pas de BourréePLOS ONE

Dear Dr. Strenge,

Thank you for submitting your manuscript to PLOS ONE. After careful consideration, we feel that it has merit but does not fully meet PLOS ONE’s publication criteria as it currently stands. Therefore, we invite you to submit a revised version of the manuscript that addresses the points raised during the review process.

We look forward to receiving your revised manuscript.

Kind regards,

Stergios Makris

Academic Editor

PLOS ONE

Journal Requirements:

“We acknowledge support for the publication costs by the Open Access Publication Fund 333 of Bielefeld University and the Deutsche Forschungsgemeinschaft (DFG).”

Reviewers' comments:

Reviewer's Responses to Questions

**Comments to the Author**

1. Is the manuscript technically sound, and do the data support the conclusions?

Reviewer #1: Yes

Reviewer #2: Partly

2. Has the statistical analysis been performed appropriately and rigorously? 

Reviewer #1: I Don't Know

Reviewer #2: Yes

3. Have the authors made all data underlying the findings in their manuscript fully available?

Reviewer #1: Yes

Reviewer #2: Yes

4. Is the manuscript presented in an intelligible fashion and written in standard English?

Reviewer #1: Yes

Reviewer #2: Yes

5. Review Comments to the Author

Reviewer #1: I appreciate the information relating cognitive skills (mental representations stored in long-term memory) to motor control. Most of the literature that I review in the area of perceptual-cognitive-motor performance lacks such detailed analysis of cognitive contributions to efficient performance of complex goal-directed actions. Despite a lack of familiarity with ballet, I clearly see the potential value of using the same approach for analysis of other sport-specific movement sequences. Concerns that I think should be addressed: no mention of Institutional Review Board approval of the study procedures, nor is there any mention of the manner in which participants were recruited/selected. There is no information pertaining to power analysis for determination of the minimum number of participants need to avoid a Type II error. I do not consider power analysis to be an essential component of the research process, but some explanation is needed for the manner in which the 18 participants were recruited (i.e., 3 perfectly balanced groups of 6 participants each).

Reviewer #2: The manuscript was well written. I have only two comments that the authors should address them in the manuscript.

1. considering the small sample size, this study should report as pilot study or preliminary study.

2. Please provide sample size estimation in the manuscript.

6. PLOS authors have the option to publish the peer review history of their article (what does this mean?). If published, this will include your full peer review and any attached files.

Reviewer #1: **Yes: **Gary B. Wilkerson, EdD, ATC

Reviewer #2: No

---

## [Author Response · Author response to Decision Letter 0]

18 Aug 2023

Academic Editor:

Dear Dr Makris,

Thank you very much for handling our manuscript and enabling us to provide a revised version. We have been committed to carefully considering all remarks to optimize the quality of the manuscript. Please find our responses to each of your remarks below.

Authors’ Response: 

Thank you for bringing this to our attention. We have revised the manuscript based on PLOS ONE's style requirements you attached and made the appropriate corrections in the manuscript. This includes corrections of the capitalization of the headings as well as the file naming for the figures. 

2. Please remove any funding-related text from the manuscript and let us know how you would like to update your Funding Statement. Please include your amended statements within your cover letter; we will change the online submission form on your behalf. 

Authors’ Response:

We have removed the paragraph from the Acknowledgments Section, and we would kindly ask you to amend this correspondingly in our online submission by adding the following sentence to the Funding Statement:

“We acknowledge support for the publication costs by the Open Access Publication Fund of Bielefeld University and the Deutsche Forschungsgemeinschaft (DFG). The funders had no role in study design, data collection and analysis, decision to publish, or preparation of the manuscript."

Authors’ Response:

Thank you for pointing this out. We inserted the full name of the ethics committee of Bielefeld University, who gave written approval to our study, in the subsection “Execution” of the “Materials and methods” section (line 154), as follows: “The study was approved by the Ethics Committee of Bielefeld University (EUB).”

4. Please include captions for your Supporting Information files at the end of your manuscript, and update any in-text citations to match accordingly.

Authors’ Response: 

Thank you for pointing this out. We have revised the captions for the supporting information files and provided in-text citations matching these captions (see lines 139 f., 152, 219 f., 339 f., 341 f.). 

5. Please review your reference list to ensure that it is complete and correct.

Authors’ Response: 

We have reviewed our reference list for completeness and correctness. During the revision process, we added three new references into the text, as well as in the reference list. These are the following references: 

(28.) Akoglu H. User’s guide to correlation coefficients. Turkish Journal of Emergency Medicine. 2018;18(3):91–93. doi:https://doi.org/10.1016/j.tjem.2018.08.001.

(29.) Mayerl J, Fröhlich M, Pieter A, Kemmler W. In: Bivariate statistische Verfahren. Berlin, Heidelberg: Springer Berlin Heidelberg; 2020. p. 57–72. Available from: https://doi.org/10.1007/978-3-662-61039-8_6.

(30.) Alsaqr AM. Remarks on the use of Pearson’s and Spearman’s correlation coefficients in assessing relationships in ophthalmic data. African Vision and Eye Health. 2021;80(1):10. doi:https://doi.org/10.4102/aveh.v80i1.612.

Reviewer 1:

Dear Dr. Wilkerson,

Thank you very much for taking the time to review our manuscript and for your appreciative and constructive comments. We found the advice helpful and understandable, and have incorporated the suggestions into our revision. We believe this improved the quality of the manuscript notably. Please find our responses to each of your remarks below. 

1. No mention of Institutional Review Board approval of the study procedures

Authors’ Response:

Thank you for pointing this out. We inserted the full name of the ethics committee of Bielefeld University, who gave written approval to our study, in the subsection “Execution” of the “Materials and methods” section (line 154), as follows:

 “The study was approved by the Ethics Committee of Bielefeld University

 (EUB).”

2. No mention of the manner in which participants were recruited/selected

Authors’ Response:

We included further information regarding the recruitment and selection of the participants in the subsection “Participants” of the “Materials and methods” section (line 137 f.), as follows:

 “In order to establish three perfectly balanced expertise groups of six

 participants each, a total of 18 participants aged between 19 and 33 years were

 recruited.”

As well as in the “Execution” of the “Materials and methods” section (line 155 f.), as follows: 

 “Participants were recruited in March 2021 at the campus of Bielefeld

 University and from a ballet school in Bielefeld (Germany), gave written

 informed consent to participate in the study…” 

3. There is no information pertaining to power analysis for determination of the minimum number of participants need to avoid a Type II error. I do not consider power analysis to be an essential component of the research process, but some explanation is needed for the manner in which the 18 participants were 

recruited (i.e., 3 perfectly balanced groups of 6 participants each).

Authors’ Response:

Thank you very much for pointing out that we had not addressed this issue in our initial submission. We have performed a Power Analysis and provide detailed information on the results in the revised subsection “Participants” of the “Materials and methods” section (lines 134 ff.), as follows: 

 “A power analysis with SPSS Version 29.0.0 estimated that a minimum

 sample size of 16 participants would have been necessary to determine

 significant Spearman correlation values of ρ = .7 or higher, which are

 commonly interpreted as “strong” [28–30], with at least 80% chance. In

 order to establish three perfectly balanced expertise groups of six

 participants each, a total of 18 participants aged between 19 and 33 years were

 recruited.”

 

Reviewer 2:

Thank you very much for taking the time to review and for your commendatory and constructive comments upon our manuscript. We found the advice helpful and understandable, and have incorporated the suggestions into our revision. We believe this improved the quality of the manuscript notably. Please find our responses to each of your remarks below. 

1. Please provide sample size estimation in the manuscript.

Authors’ Response:

Thank you very much for pointing out that we had not addressed this issue in our initial submission. We have performed a Power Analysis and provide detailed information on the results in the revised subsection “Participants” of the “Materials and methods” section (lines 134 ff.), as follows: 

 “A power analysis with SPSS Version 29.0.0 estimated that a minimum

 sample size of 16 participants would have been necessary to determine

 significant Spearman correlation values of ρ = .7 or higher, which are

 commonly interpreted as “strong” [28–30], with at least 80% chance. In

 order to establish three perfectly balanced expertise groups of six

 participants each, a total of 18 participants aged between 19 and 33 years were

 recruited.”

2. considering the small sample size, this study should report as pilot study or preliminary study.

Authors’ Response:

We understand that the sample size, which was attained under constraints of the COVID-19 pandemic, may appear comparatively small. However, due to the fact that this participant number already exceeded the minimum sample size estimated by power analysis and yielded highly significant results, we tend to not classify the study as a mere pilot or preliminary study. Nevertheless, we agree that the sample size needs to be considered in interpreting the results of our study. Therefore, we have added a corresponding additional remark in the limitations of our study (lines 296 ff.): 

„Although the participant number exceeded the minimum sample size estimated by power analysis, future larger studies could facilitate the determination of more reliable estimates of the actual population values.”

---

## [Decision Letter · Decision Letter 1]

13 Sep 2023

Expertise-dependent differences in mental representation metrics of pas de bourrée

PONE-D-23-11698R1

Dear Dr. Strenge,

We’re pleased to inform you that your manuscript has been judged scientifically suitable for publication and will be formally accepted for publication once it meets all outstanding technical requirements.

Kind regards,

Stergios Makris

Academic Editor

PLOS ONE

Additional Editor Comments (optional):

Reviewers' comments:

Reviewer's Responses to Questions

**Comments to the Author**

1. If the authors have adequately addressed your comments raised in a previous round of review and you feel that this manuscript is now acceptable for publication, you may indicate that here to bypass the “Comments to the Author” section, enter your conflict of interest statement in the “Confidential to Editor” section, and submit your "Accept" recommendation.

Reviewer #1: (No Response)

Reviewer #2: All comments have been addressed

2. Is the manuscript technically sound, and do the data support the conclusions?

Reviewer #1: Yes

Reviewer #2: Yes

3. Has the statistical analysis been performed appropriately and rigorously? 

Reviewer #1: Yes

Reviewer #2: Yes

4. Have the authors made all data underlying the findings in their manuscript fully available?

Reviewer #1: Yes

Reviewer #2: Yes

5. Is the manuscript presented in an intelligible fashion and written in standard English?

Reviewer #1: Yes

Reviewer #2: Yes

6. Review Comments to the Author

Reviewer #1: After a thorough comparison of the original manuscript content with that of the revised version, I have no further concerns about the work's suitability for publication.

Reviewer #2: (No Response)

7. PLOS authors have the option to publish the peer review history of their article (what does this mean?). If published, this will include your full peer review and any attached files.

Reviewer #1: **Yes: **Gary B. Wilkerson, EdD, ATC

Reviewer #2: No

---

## [Editor Report · Acceptance letter]

25 Sep 2023

PONE-D-23-11698R1 

Expertise-dependent differences in mental representation
metrics of pas de bourrée 

Dear Dr. Strenge:

I'm pleased to inform you that your manuscript has been deemed suitable for publication in PLOS ONE. Congratulations! Your manuscript is now with our production department. 

Kind regards, 

on behalf of

Dr. Stergios Makris 

Academic Editor

PLOS ONE